# MEMORY-CONSTRAINED POLICY OPTIMIZATION

## ABSTRACT

We introduce a new constrained optimization method for policy gradient reinforcement learning, which uses *two trust regions* to regulate each policy update. In addition to using the proximity of one single old policy as the first trust region as done by prior works, we propose to form a second trust region through the construction of another virtual policy that represents a wide range of past policies. We then enforce the new policy to stay closer to the virtual policy, which is beneficial in case the old policy performs badly. More importantly, we propose a mechanism to automatically build the virtual policy from a memory buffer of past policies, providing a new capability for dynamically selecting appropriate trust regions during the optimization process. Our proposed method, dubbed as Memory-Constrained Policy Optimization (MCPO), is examined on a diverse suite of environments including robotic locomotion control, navigation with sparse rewards and Atari games, consistently demonstrating competitive performance against recent on-policy constrained policy gradient methods.

## 1  INTRODUCTION

Reinforcement learning (RL) combined with neural networks is the current workhorse in machine learning. Using neural networks to approximate value and policy functions enables classical approaches such as Q-learning and policy gradient to achieve promising results on many challenging problems such as Go, Atari games and robotics (Silver et al., 2017; Mnih et al., 2015; Lillicrap et al., 2016; Mnih et al., 2016). Compared to Deep Q-learning, deep policy gradient (PG) methods are often more flexible and applicable to both discrete and continuous action problems. However, these methods tend to suffer from high sample complexity and training instability since the gradient may not accurately reflect the policy gain when the policy changes substantially (Kakade & Langford, 2002). This is exacerbated for deep policy networks where numerous parameters need to be optimized and small updates in parameter space can lead to huge changes in policy space.

To address this issue, one solution is to regularize each policy update by restricting the Kullback–Leibler (KL) divergence between the new policy and the previous one, which can guarantee monotonic policy improvement (Schulman et al., 2015a). However, jointly optimizing the approximate advantage function and the KL term does not work in practice. Therefore, Schulman et al. (2015a) proposed Trust Region Policy Optimization (TRPO) to constrain the new policy within a KL divergence radius, which requires second-order gradients. Alternatives such as Proximal Policy Optimization (PPO) (Schulman et al., 2017) use a simpler first-order optimization with adaptive KL or clipped surrogate objective while still maintaining the reliable performance of TRPO. Recent methods recast the problem through a new lens using Expectation-Maximization or Mirror Descent Optimization, and this also results in first-order optimization with KL divergence term in the loss function (Abdolmaleki et al., 2018; Song et al., 2019; Yang et al., 2019; Tomar et al., 2020).

An issue with the above methods is that the previous policy used to restrict the new policy may be suboptimal and thus unreliable in practice. For example, due to stochasticity and imperfect approximations, consider that the new policy may fall into a local optimum even under trust-region optimizations. Then in the next update, this policy will become the "previous" policy, and will continue pulling the next policy to stay in the local optimum, thus slowing down the training progress. For on-policy methods using mini-batch updates like PPO, the situation is more complicated as the "previous" policy is defined as the old policy that was used to collect data, which can be either very far or close to the current policy. There is no guarantee that the old policy defines a reasonable trust region for regulating the new policy.

In this paper, we propose a novel constrained policy iteration procedure, dubbed as Memory-Constrained Policy Optimization (MCPO), wherein a *virtual policy* representing past policies takes part in regularizing each policy update. The virtual and the old policy together form *two trust regions* that attract the new policy by minimizing two KL divergences. The virtual policy is designed to complement the old policy, attracting the new policy more when the old policy performs badly. As such, we assign different coefficient weights to the two KL terms. The coefficient weights are computed dynamically based on the performance of the two policies (the higher performer yields higher coefficient weights). To create the virtual policy, we maintain a memory buffer of past policies, from which we build a mixture of policies. In particular, we use a neural network–named the *attention network*, which takes the context surrounding the current, the old and the virtual policy, to generate the attention weights to each policy in the memory. Then, we use the attention weights to perform a convex combination of the parameters of each policy, forming the virtual policy. The attention network is optimized to maximize the approximate expected advantages of the virtual policy. To train our system, we jointly optimize the policy and attention networks, alternating between sampling data from the policy and updating the networks in a mini-batch manner.

We verify our proposed MCPO through a diverse set of experiments and compare ours with the performance of recent constrained policy optimization baselines. In our experiment on classical control tasks, amongst tested models, MCPO shows minimal sensitivity to hyperparameter changes, consistently achieving good performance across tasks and hyperparameters. Our testbed on 6 Mujoco tasks shows that MCPO with a big policy memory consistently outperforms others where the attention network plays an important role. We also demonstrate MCPO's capability of learning efficiently on sparse reward and high-dimensional problems such as navigation and Atari games. Finally, our ablation study highlights the necessity of each component in MCPO.

## 2    BACKGROUND: POLICY OPTIMIZATION WITH TRUST REGION

In this section, we briefly review some fundamental constrained policy optimization approaches. A general idea is to force the new policy $\pi_\theta$ to be close to a recent policy $\pi_{\theta_{old}}$. In this paper, we refer to a policy as its parameters (i.e. policy $\theta$ means policy $\pi_\theta$).

**Conservative Policy Iteration (CPI)** The method starts with a basic objective of policy gradient algorithms, which is to maximize the expected advantage $\hat{A}_t$.

$$L^{CPI}(\theta) = \hat{\mathbb{E}}_t \left[ \frac{\pi_\theta(a_t|s_t)}{\pi_{\theta_{old}}(a_t|s_t)} \hat{A}_t \right]$$

where the advantage $\hat{A}_t$ is a function of returns collected from $(s_t, a_t)$ by using $\pi_{\theta_{old}}$ (see Appendix A.2) and $\hat{\mathbb{E}}_t[\cdot]$ indicates the empirical average over a finite batch of data. To constrain policy updates, the new policy is a mixture of the old and the greedy policy: $\tilde{\theta} = \arg\max L^{CPI}(\theta)$. That is, $\theta = \alpha\theta_{old} + (1 - \alpha)\tilde{\theta}$ where $\alpha$ is the mixture hyperparameter (Kakade & Langford, 2002). As the data is sampled from previous iteration's policy $\theta_{old}$, the objective needs importance sampling estimation. Hereafter, we denote $\frac{\pi_\theta(a_t|s_t)}{\pi_{\theta_{old}}(a_t|s_t)}$ as $rat_t(\theta)$ for short.

**KL-Regularized Policy Optimization (with fixed or adaptive KL coefficient)** Another way to enforce the constraint is to jointly maximize the expected advantage and minimize KL divergence between the new and old policy, which ensures monotonic improvement (Schulman et al., 2015a).

$$L^{KLPO}(\theta) = \hat{\mathbb{E}}_t \left[ rat_t(\theta) \hat{A}_t - \beta KL \left[ \pi_{\theta_{old}}(\cdot|s_t), \pi_\theta(\cdot|s_t) \right] \right]$$

where $\beta$ is a hyperparameter that controls the degree of update conservativeness, which can be fixed (KL Fixed) or changed (KL Adaptive) during training (Schulman et al., 2017).

**Trust Region Policy Optimization (TRPO)** The method optimizes the expected advantage with hard constraint (Schulman et al., 2015a). This is claimed as a practical implementation that is less conservative than the theoretically justified algorithm using KL regularizer mentioned above.

$$L^{TRPO}(\theta) = \hat{\mathbb{E}}_t \left[ rat_t(\theta) \hat{A}_t \right]$$
$$\text{st } \delta \geq KL\left[ \pi_{\theta_{old}}(\cdot|s_t), \pi_\theta(\cdot|s_t) \right]$$

where $\delta$ is the KL constraint radius.

**Proximal Policy Optimization (PPO)** PPO is a family of constrained policy optimization, which uses first-order optimization and mini-batch updates including KL Adaptive and clipped PPO. In this paper, we use PPO to refer to the method that limits the change in policy by clipping the loss function (clipped PPO) (Schulman et al., 2017).

$$L^{PPO}(\theta) = \hat{\mathbb{E}}_t \left[ \min\left( rat_t(\theta)\hat{A}_t, \text{clip}\left( rat_t(\theta), 1-\epsilon, 1+\epsilon \right) \hat{A}_t \right) \right]$$

where $\epsilon$ is the clip hyperparameter.

In the above equations, $\theta$ is the currently optimized policy, which is also referred to as the current policy. $\theta_{old}$ represents a past policy, which can be one step before the current policy or the last policy used to interact with the environment. In either case, the rule to decide $\theta_{old}$ is fixed throughout training. If for some reason, $\theta_{old}$ is not optimal, it is unavoidable that the following updates will be negatively impacted. We will address this issue in the next section.

## 3 MEMORY-CONSTRAINED POLICY OPTIMIZATION (MCPO)

### 3.1 TWO KL DIVERGENCE CONSTRAINTS

In trust-region methods with mini-batch updates such as PPO, the old policy $\theta_{old}$ is often chosen as the last policy that is used to collect observations from the environment. Before the next environment interaction, this old policy is fixed across policy updates, and can be one or many steps before the current policy depending on the mini-batch size and the number of update loops. This can be detrimental to the optimization if this old policy is poor in quality, forcing the following updates to be close to a poor solution. *To tackle this issue, we propose to constrain the new policy not only to the policy $\theta_{old}$, but also to a changeable policy that is representative of many past policies.* Let $\psi$ denote the *virtual policy* that represents the history of policies. $\psi$ is dynamically computed based on the past policies using attention mechanism (see Sec. 3.2). In contrast to the fixed $\theta_{old}$, depending on the attention weights, $\psi$ can represent a further or closer checkpoint to the current policy than $\theta_{old}$. We use both $\psi$ and $\theta_{old}$ to construct the objective function as follows,

$$
\begin{aligned}
L_1(\theta) = & \hat{\mathbb{E}}_t \left[ rat_t(\theta) \hat{A}_t \right] \\
& - \beta \hat{\mathbb{E}}_t \left[ (1 - \alpha_t(\cdot|s_t)) KL\left[ \pi_{\theta_{old}}(\cdot|s_t), \pi_\theta(\cdot|s_t) \right] \right] \\
& - \beta \hat{\mathbb{E}}_t \left[ \alpha_t(\cdot|s_t) KL\left[ \pi_\psi(\cdot|s_t), \pi_\theta(\cdot|s_t) \right] \right]
\end{aligned}
\tag{1}
$$

where $\alpha_t(\cdot|s_t)$ is the coefficient weight resembling a forget gate, and $\beta$ is the scaling coefficient of the KL constraint terms. In this paper, the expectation is estimated by taking average over $t$ in a mini-batch of sampled data.

The forget gate determines how much the new policy should forget the virtual policy from the memory and focus on the $\theta_{old}$. Intuitively, if the virtual policy is better than the old policy, the new policy should be kept close to the virtual policy and vice versa. Hence, $\alpha_t = \frac{e^{R_t(\psi)}}{e^{R_t(\psi)} + e^{R_t(\theta_{old})}}$ where $R_t(\psi)$ measures the performance of the policy $\psi$, which can be estimated by weighted importance sampling. That is $R_t(\psi) = rat_t(\psi)\hat{A}_t$.

Besides deciding which trust region the new policy should rely on, we dynamically weigh the whole KL terms via adjusting $\beta$. Using a fixed threshold $d_{targ}$ to change $\beta$ (e.g in KL Adaptive, if $KL\left[ \pi_{\theta_{old}}(\cdot|s_t), \pi_\theta(\cdot|s_t) \right] > d_{targ}$, increase $\beta$ (Schulman et al., 2017)) showed limited performance since $d_{targ}$ should vary depending on the current learning. We instead make use of $\psi$ as a

reference for selecting $\beta$. Let $d(a, b) = \hat{\mathbb{E}}_t [KL [\pi_a (\cdot|s_t), \pi_b (\cdot|s_t)]]$ denote the "distance" between 2 policies $\pi_a$ and $\pi_b$, we propose a *switching-$\beta$* rule as follows,

$$\begin{cases} \beta = \beta_{max} & \text{if } d(\theta_{old}, \theta) > d(\theta_{old}, \psi) \\ \beta = \beta_{min} & \text{otherwise} \end{cases} \tag{2}$$

The intuition is that we encourage the enforcement of the constraint when $\pi_\theta$ is too far from $\pi_{\theta_{old}}$ using the distance between $\pi_{\theta_{old}}$ and $\pi_\psi$ as a reference.

---

**Algorithm 1** Memory-Constrained Policy Optimization (for 1 actor).

---

**Require:** A policy buffer $\mathcal{M}$, an initial policy $\pi_{\theta_{old}}$. $T, K, B$ are the learning horizon, number of update epochs, and batch size, respectively.
1: Initialize $\psi_{old} \leftarrow \theta_{old}, \theta \leftarrow \theta_{old}$
2: **for** $iteration = 1, 2, ...$ **do**
3:     Run policy $\pi_{\theta_{old}}$ in environment for $T$ timesteps. Compute advantage estimates $\hat{A}_1, ..., \hat{A}_T$
4:     **for** $epoch = 1, 2, ...K$ **do**
5:       **for** $batch = 1, 2, ...T/B$ **do**
6:         Compute $\psi$ (Eq. 4) using $\psi_{old}, \theta, \theta_{old}$, then optimize $L^{MCPO}$ wrt $\theta$ (Eq. 5)
7:         **if** $d(\theta, \psi) > d(\theta_{old}, \psi)$ **then** add $\theta$ to $\mathcal{M}$
8:         **if** $|\mathcal{M}| > N$ **then** remove the last item from $\mathcal{M}$
9:         $\psi_{old} \leftarrow \psi$
10:       **end for**
11:     **end for**
12:     $\theta_{old} \leftarrow \theta$
13: **end for**

---

### 3.2 LEARNING TO GENERATE THE VIRTUAL POLICY

It is critical to compute a suitable virtual policy. On one hand, if the virtual policy is too far from the currently optimizing point, it will be irrelevant, pulling the new policy back and postponing the learning. On the other hand, if the virtual policy is too recent, it will not complement $\theta_{old}$ and cannot prevent major changes in the policy update. Also, it is reasonable to find a virtual policy that has good performance on current data. Otherwise, using trust regions near poor policies could destroy the learning. We will utilize these intuitions to build the virtual policy.

**Policy Memory** We first maintain a memory buffer $\mathcal{M}$ that stores past policy parameters $\mathcal{M} = \{\theta_j\}_{j=1}^{|\mathcal{M}|}$. During optimization, we add a policy's parameter to $\mathcal{M}$ if it is far enough from $\psi$. In particular, we measure the distances $d(\theta_{old}, \psi)$ and $d(\theta, \psi)$, then propose *conditional writing*:

$$\text{Add } \theta \text{ to } \mathcal{M} \text{ if } d(\theta, \psi) > d(\theta_{old}, \psi) \tag{3}$$

The memory capacity is $N$. When the memory is full, we discard the earliest policy in $\mathcal{M}$.

**Context Vector** We hypothesize that the context surrounding $\theta_{old}, \psi_{old}$ and $\theta$, where $\psi_{old}$ is the last virtual policy, plays a role in determining the next virtual policy. We build the context by extracting specific features: pair-wise distances between policies ($d(\theta_{old}, \psi_{old})$, $d(\theta, \psi_{old})$, $d(\theta_{old}, \theta)$), the empirical returns of these policies ($\hat{\mathbb{E}}_t [R_t (\psi_{old})]$, $\hat{\mathbb{E}}_t [R_t (\theta)]$, $\hat{\mathbb{E}}_t [R_t (\theta_{old})]$), policy entropy ($\hat{\mathbb{E}}_t [-\log (\pi_{\psi_{old}} (\cdot|s_t))]$, $\hat{\mathbb{E}}_t [-\log (\pi_\theta (\cdot|s_t))]$, $\hat{\mathbb{E}}_t [-\log (\pi_{\theta_{old}} (\cdot|s_t))]$) and value losses ($\hat{\mathbb{E}}_t (V_{\psi_{old}} (s_t) - V_{target} (s_t))^2$, $\hat{\mathbb{E}}_t (V_\theta (s_t) - V_{target} (s_t))^2$, $\hat{\mathbb{E}}_t (V_{\theta_{old}} (s_t) - V_{target} (s_t))^2$). These features form a *context vector* $v_{context}$ that captures the properties of each policy and the relationship between them, which can represent the context that generates the attention weights.

**Attention Mechanism** We argue that the virtual policy should be determined based on the context. A simple strategy such as taking average of policies in $\mathcal{M}$ is likely sub-optimal as the quality of these policies vary and some can be irrelevant to the current learning context. Hence, we propose to sum the policies in a weighted manner wherein the weights are generated by a neural network whose input is $v_{context}$. We compute $\psi$ by performing "attention" in the parameter space as follows,

| Model | Pendulum 1M | LunarLander 1M | BWalker 5M |
|---|---|---|---|
| KL Adaptive ($d_{targ} = 0.003$) | -407.74±484.16 | 238.30±34.07 | 206.99±5.34 |
| KL Adaptive ($d_{targ} = 0.01$) | -147.52±9.90 | *254.26±19.43* | 247.70±14.16 |
| KL Adaptive ($d_{targ} = 0.03$) | -601.09±273.18 | 246.93±12.57 | *259.80±6.33* |
| KL Fixed ($\beta = 0.01$) | -1051.14±158.81 | 247.61±19.79 | 221.55±38.64 |
| KL Fixed ($\beta = 0.1$) | -464.29±426.27 | *256.75±20.53* | *263.56±10.04* |
| KL Fixed ($\beta = 1$) | *-136.40±4.49* | 192.62±32.97 | 215.13±13.29 |
| PPO (clip $\epsilon = 0.1$) | -282.20±243.42 | 242.98±13.50 | 205.07±19.13 |
| PPO (clip $\epsilon = 0.2$) | -514.28±385.34 | *256.88±20.33* | 253.58±7.49 |
| PPO (clip $\epsilon = 0.3$) | -591.31±229.32 | *259.93±22.52* | *260.51±17.86* |
| MDPO ($\beta_0 = 0.5$) | *-136.45±8.21* | 247.96±4.74 | 251.18±29.10 |
| MDPO ($\beta_0 = 1$) | *-139.14±10.32* | 207.96±43.86 | 245.27±10.47 |
| MDPO ($\beta_0 = 2$) | *-135.52±5.28* | 227.76±16.96 | 226.80±15.67 |
| VMPO ($\alpha_0 = 0.1$) | *-144.51±7.04* | 201.87±29.48 | 236.57±10.62 |
| VMPO ($\alpha_0 = 1$) | *-139.50±5.54* | 212.85±43.35 | 238.82±11.11 |
| VMPO ($\alpha_0 = 5$) | *-296.48±213.06* | 222.13±35.55 | 164.40±40.36 |
| MCPO ($N = 5$) | **-133.42±4.53** | *262.23±12.47* | *265.80±5.55* |
| MCPO ($N = 10$) | -146.88±3.78 | *263.04±11.48* | **266.26±8.87** |
| MCPO ($N = 40$) | *-135.57±5.22* | **267.19±13.42** | 249.51±12.75 |

Table 1: Mean and std. over 5 runs on classical control tasks (with number of training environment steps). Bold denotes the best mean. Underline denotes good results (if exist), statistically indifferent from the best in terms of Cohen effect size less than 0.5.

$$\psi = \sum_{j}^{|\mathcal{M}|} f_{\varphi}\left(v_{context}\right)_j \theta_j \tag{4}$$

where $f_{\varphi}$ is the attention network– a feed-forward neural network parameterized by $\varphi$ with $\mathrm{softmax}$ activation. The network outputs a $N$-dimensional output vector establishing the attention weights. Here $\psi$ is a function of $\varphi$ and we train $\varphi$ to improve $\psi$'s performance.

**Objective function** Given the virtual policy $\psi\left(\varphi\right)$, we find $\varphi$ to maximize its performance $L_2\left(\varphi\right) = \hat{\mathbb{E}}_t\left[R_t\left(\psi\left(\varphi\right)\right)\right]$. We aim to obtain the best representative of past policies without examining the performance of each individual policy in $\mathcal{M}$ since evaluating all policies is highly expensive, especially when $|\mathcal{M}|$ is large. In addition, learning a "soft" attention is more flexible than searching for a "hard" policy that performs best (in terms of $L_2$) since the performance measurement itself can be noisy and not always reliable. To train the whole system, we use gradient ascent to maximize

$$L^{MCPO} = L_1\left(\theta\right) + L_2\left(\varphi\right). \tag{5}$$

When optimizing $L_1$, we fix $\varphi$ and only update $\theta$ to avoid gradient back-propagation via $\varphi$ (similarly, when optimizing $L_2$, we fix $\theta$ and only update $\varphi$). Theoretical motivation for the design of $L_1$ and $L_2$ is given in Appendix C. We implement MCPO using minibatch update procedure (Schulman et al., 2017). Algo. 1 illustrates a high-level implementation of MCPO with 1 actor.

## 4 EXPERIMENTAL RESULTS

In our experiments, the main baselines are recent on-policy constrained methods that use first-order optimization, in which most of them employ KL terms in the objective function. They are KL Adaptive, KL Fixed, PPO (Schulman et al., 2017), MDPO (Tomar et al., 2020) and VMPO (Song et al., 2019). We also include second-order methods such as TRPO (Schulman et al., 2015a) and ACKTR (Wu et al., 2017). Across experiments, for MCPO, we fix $\beta_{max} = 10$, $\beta_{min} = 0.01$ and only tune $N$. More details on the baselines and tasks are given in Appendix B.1.

Figure 1: Unlock (left) and UnlockPickup (right)'s learning curves (mean and std. over 10 runs).

## 4.1 CLASSICAL CONTROL: HYPERPARAMETER SENSITIVITY TEST

In this section, we compare MCPO to other first-order policy gradient methods (KL Adaptive, KL Fixed, PPO, MDPO and VMPO) on 3 classical control tasks: Pendulum, LunarLander and Bipedal-Walker, which are trained for one, one and five million environment steps, respectively. Here, we are curious to know whether tuning hyperparameters helps the baselines solve these simple tasks, and how their performances fluctuate as the hyperparameters vary. For each model, we choose one hyperparameter that controls the conservativeness of policy update, and we try different values for the signature hyperparameter while keeping the others the same. For example, for PPO, we tune the clip value $\epsilon$; for KL Fixed we tune $\beta$ coefficient and these possible values are chosen following prior works. For our MCPO, we tune the size of the policy memory $N$ (5, 10 and 40). We do not try bigger policy memory size to keep MCPO running efficiently (see Appendix B.2 for details of baselines, the choice of hyperparameters and running time analysis).

Table 1 reports the results of MCPO and 5 baselines with different hyperparameters. For these simple tasks, tuning the hyperparameters often helps the model achieve at least moderate performance. However, models like KL Adaptive and VMPO cannot reach good performance despite being tuned. PPO shows good results on LunarLander and BipedalWalker, yet underperfoms others on Pendulum. Interestingly, if tuned properly, the vanilla KL Fixed can show competitive results compared to PPO and MDPO in BipedalWalker. Amongst all, our MCPO with suitable $N$ achieves the best performance on all tasks. Remarkably, its performance does not fluctuate much as $N$ changes from 5 to 40, often obtaining good and best results. On the contrary, other methods observe clear drop in performance if the hyperparameters are set incorrectly (see Appendix B.2 for full learning curves).

## 4.2 NAVIGATION TASKS: SAMPLE EFFICIENCY TEST

Here, we validate our method on sparse reward environments using MiniGrid library (Chevalier-Boisvert et al., 2018). In particular, we test MCPO and other baselines (same as above) on Unlock and UnlockPickup tasks. In these tasks, the agent navigates through rooms and picks up objects to complete the episode. The agent only receives reward +1 if it can complete the episode successfully. For sample efficiency test, we train all models on Unlock (find key and open the door) and Unlock-Pickup (find key, open the door and pickup an object), for only 100,000 and 1 million environment steps, respectively. The models use the best conservative hyperparameters found in the previous task (more in Appendix B.3).

Fig. 1 shows the learning curves of examined models on these two tasks. For Unlock task, except for MCPO and VMPO, 100,000 steps seem not enough for the models to learn useful policies. When trained with 1 million steps on UnlockPickup, the baselines can find better policies, yet still underpefrom MCPO. Here VMPO shows faster learning progress than MCPO at the beginning, however it fails to converge to the best solution. Our MCPO is the best performer, consistently ending up with average return of 0.9 (90% of episodes finished successfully).

To illustrate how the virtual policy supports building trust regions to boost MCPO's performance, we analyze the relationships amongst the old ($\theta$), the virtual policy ($\psi$) and the policies stored in $\mathcal{M}$ ($\theta_j$) throughout Unlock training. Fig. 2 (a) plots the location of these policies over a truncated period of training (from update step 5160 to 7070). Due to conditional writing rule, the steps where policies are added to $\mathcal{M}$ can be uneven (first row-red lines), often distributed right after the locations of the old policy (second row-green lines). We query at 10-th step behind the old policy (fourth row-cyan lines) to find which policy in $\mathcal{M}$ has the highest attention (third row-yellow lines, linked by blue arrows). As shown in Fig. 2 (a) (second and third row), the attended policy, which mostly

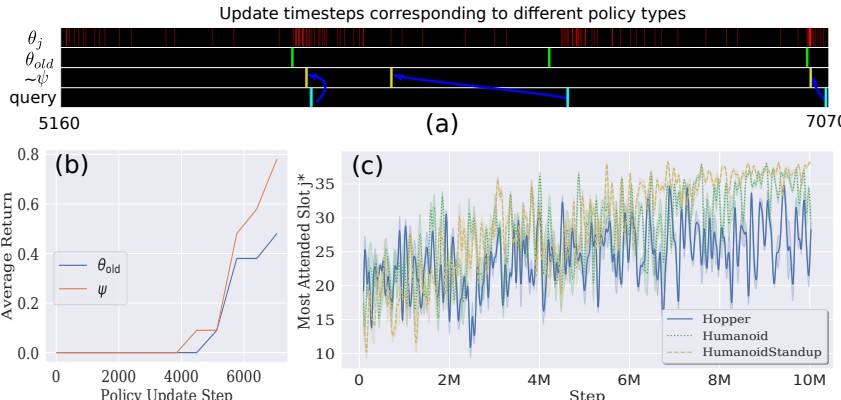

Figure 2: (a) Policy analysis on Unlock. First row (red lines): steps where a policy is added to $\mathcal{M}$, i.e. the steps of $\theta_j$. Second row (green lines): steps of old policies $\theta_{old}$. Third row (yellow lines): steps of mostly attended policy, approximating $\psi$. Fourth row (cyan lines): 3 steps of interest where we want to find their attended steps. Blue arrows link a query step and the step that receives highest attention. (b) Quality of $\psi$ vs. $\theta_{old}$. Average return collected by $\psi$ and $\theta_{old}$ at different stages of training. (c) 3 Mujoco tasks. The slot in $\mathcal{M}$ received the highest attention $j^* = \mathrm{argmax}_j \, f_\varphi \left(v_{context}\right)_j$ over time.

resembles $\psi$, can be further or closer to the query step than the old policy depending on the training stage. Since we let the attention network learn to attend to the policy that maximizes the advantage of current mini-batch data, the attended one is not necessarily the same as the old policy.

The choice of the chosen virtual policy being better than the old policy is shown in Fig. 2 (b) where we collect several checkpoints of virtual and old policies across training and evaluate each of them on 10 testing episodes. Here using $\psi$ to form the second KL constraint is beneficial as the new policy is generally pulled toward a better policy during training. That contributes to the excellent performance of MCPO compared to other single trust-region baselines, especially KL Fixed and Adaptive, which are very close to MCPO in term of objective function style.

### 4.3 MUJOCO CONTINUOUS CONTROL: EFFECTIVENESS OF LEARNED $\psi$

Next, we examine MCPO and some trust-region methods from the literature that are known to have good performance on continuous control problems: TRPO, PPO and MDPO. To understand the role of the attention network in MCPO, we design a variant of MCPO: *Mean $\psi$*, which simply constructs $\psi$ by taking average over policy parameters in $\mathcal{M}$. We pick 6 hard Mujoco tasks and train each model for 10 million environment steps. For each baseline, we again only tune the conservative hyperparameters and report the best configurations in Table 2 (see Appendix B.4 for full results).

The results show that MCPO is the best performer on 5 out of 6 tasks, where clear outperformance gaps can be found in HalfCheetah, Ant, Humanoid and HumanoidStandup. We note that this is only achieved as MCPO uses $N = 40$, which indicates that bigger policy memory (more conservativeness) is beneficial in this case. The variant Mean $\psi$ demonstrates reasonable performance for the first 4 tasks, yet almost fails to learn on the last two, which means using a mean virtual policy is unsuitable in these tasks.

To understand the effectiveness of the attention network, we visualize the attention pattern of MCPO on the last two tasks and on Hopper-a task that Mean $\psi$ performs well. Fig. 2 (c) illustrates that for the first two harder tasks, MCPO gradually learns to favour older policies in $\mathcal{M}$ ($j^* > 35$), which puts more restriction on the policy change as the model converges. This strategy seems critical for those tasks as the difference in average return between learned $\psi$ and Mean $\psi$ is huge in these cases. On the other hand, on Hopper, the top attended slots are just above the middle policies in $\mathcal{M}$ ($j^* \sim 25$), which means this task prefers an average restriction.

| Model | HalfCheetah | Walker2d | Hopper | Ant | Humanoid | HumanoidStandup |
|---|---|---|---|---|---|---|
| TRPO | 2,811±114 | 3,966±56 | 3,159±72 | 2,438±402 | 4,576±106 | 145,143±3,702 |
| PPO | 4,753±1,614 | **5,278±594** | 2,968±1,002 | 3,421±534 | 3,375±1,684 | 155,494±6,663 |
| MDPO | 4,774±1,598 | 4,957±330 | 3,153±956 | 3,553±696 | 1,620±2,145 | 90,646±5,855 |
| Mean $\psi$ | 4,942±3,095 | 5,056±842 | 3,430±259 | *4,570±548* | 353±27 | 71,308±11,113 |
| MCPO | **6,173±595** | *5,120±588* | ***3,620±252*** | **4,673±249** | **4,848±711** | **195,404±32,801** |

Table 2: Mean and std. over 5 runs on 6 Mujoco tasks at 10M environment steps.

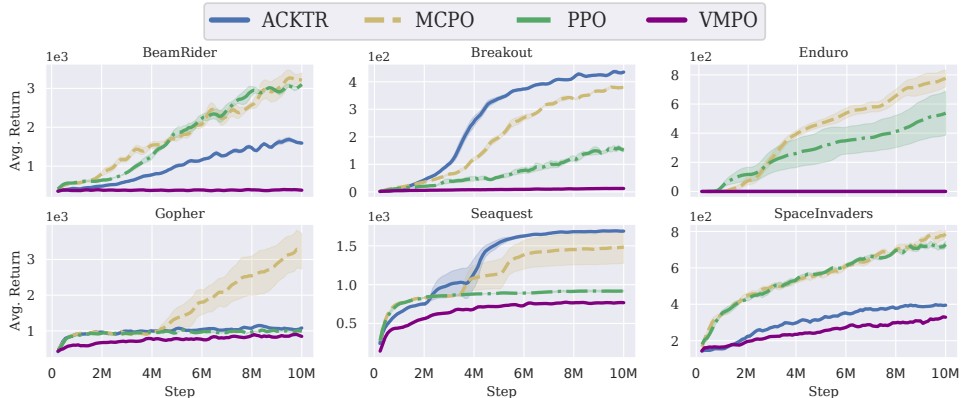

Figure 3: Mean and std. over 5 runs on 6 Atari games over 10M environment steps.

## 4.4 ATARI GAMES: SCALING TO HIGHER DIMENSIONS

As showcasing the robustness of our method to high-dimensional inputs, we execute an experiment on a subset of Atari games wherein the states are screen images and the policy and value function approximator uses deep convolutional neural networks. We choose 6 typical games (Mnih et al., 2013) and benchmark MCPO against PPO, ACKTR and VMPO, training all models for only 10 million environment steps. In this experiment, MCPO uses $N = 10$ and other baselines' hyperparameters are selected based on the original papers (see Appendix B.5).

Fig. 3 visualizes the learning curves of the models. Regardless of our regular tuning, VMPO performs poorly, indicating that this method is unsuitable or needs extensive tuning to work for low-sample training regime. ACKTR, works extremely well on certain games (Breakout and Seaquest), but shows mediocre results on others (Enduro, BeamRider), overall underperfoming MCPO. PPO is always similar or inferior to MCPO on this testbed. Our MCPO always demonstrates competitive results, outperforming all other models in 4 games, especially on Enduro and Gopher, and showing comparable results with that of the best model in the other 2 games.

## 4.5 ABLATION STUDY

Finally, we verify MCPO's 3 components: virtual policy (Eq. 1), switching-$\beta$ (Eq. 2) and conditional writing rule (Eq. 3). We also confirm the role of choosing the right memory size $N$ and learning to attend to the virtual policy $\psi$. As such, we pick BipedalWalkerHardcore from OpenAI Gym and train MCPO with different configurations for 50M steps. First, we tune $N$ (5,10 and 40) using the normal MCPO with all components on and find out that $N = 10$ is the best, reaching about 169 return score. Keeping $N = 10$, for each component, we ablate or replace our component with an alternative and report the findings as follows.

**Virtual policy** To show the benefit of pushing the new policy toward the virtual policy, we implement a variant of MCPO that does not use $\psi$'s KL term in Eq. 1 (a.k.a. $\alpha_t = 0$). This variant underperfoms the normal MCPO by a margin of 100 return. **Switching-$\beta$** The results show that compared to the annealed $\beta$ strategy adopted from MDPO, our switching-$\beta$ achieves significantly better results with about 50 return score higher. **Conditional writing** We compare our proposal with the vanilla approach that adds a new policy to $\mathcal{M}$ at every update step (frequent writing) and another

version that writes to $\mathcal{M}$ every interval of 10 update steps (uniform writing). Both frequent and uniform writing show slow learning progress, and ends up with negative rewards. Perhaps, frequently adding policies to $\mathcal{M}$ makes the memory content similar, hastening the removal of older, yet maybe valuable policies. Uniform writing is better, yet it can still add similar policies to $\mathcal{M}$ and requires additional effort for tuning the writing interval. **Learned** $\psi$ To benchmark, we try alternatives: (1) using Mean $\psi$ and (2) only using half of the features in $v_{context}$ to generate $\psi$ (Eq. 4). The results confirm that the Mean $\psi$ is not a strong baseline for this task, only reaping moderate rewards. Using less features for the context causes information loss, hinder generations of useful $\psi$, and thus, underperforms the full-feature version by a margin of 50 return. For completeness, we also compare our methods to heavily tuned baselines and confirm that the normal MCPO ($N = 10$) is the best performer. All the learning curves and details can be found in Appendix B.6.

## 5 RELATED WORKS

A framework for model-free reinforcement learning with policy gradient is approximate policy iteration (API), which alternates between estimating the advantage of the current policy, and updating the current policy's parameters to obtain a new policy by maximizing the expected advantage (Bertsekas & Tsitsiklis, 1995; Sutton et al., 2000). Theoretical studies have shown that constraining policy updates is critical for API (Kakade & Langford, 2002; Schulman et al., 2015a; Shani et al., 2020; Vieillard et al., 2020b). An early attempt to improve API is Conservative Policy Iteration (CPI), which sets the new policy as a stochastic mixture of previous greedy policies (Kakade & Langford, 2002). The mixture idea of CPI inspires other methods which explore different ways to estimate the greedy policy (Pirotta et al., 2013) or employs neural networks as function approximators (Vieillard et al., 2020a). Our paper differs from these works in three aspects: (1) we do not directly set the new policy to the mixture, rather, we use a mixture of previously found policies (the virtual policy) to define the trust region constraining the new policy via KL regularization; (2) our mixture can consist of more than 2 modes, and thus using multiple mixture weights (attention weights); (3) we use the attention network to learn these weights based on the training context.

Also motivated by Kakade & Langford (2002), TRPO extends the theory to general stochastic policies, rather than just mixture polices, ensuring monotonic improvement by combining maximizing the approximate expected advantage with minimizing the KL divergence between two consecutive policies (Schulman et al., 2015a). Arguing that optimizing this way is too conservative and hard to tune, the authors reformulate the objective as a constrained optimization problem to solve it with conjugate gradient and line search. To simplify the implementation of TRPO, Schulman et al. (2017) introduces first-order optimization methods and code-level improvement, which results in PPO–an API method that optimizes a clipped surrogate objective using minibatch updates.

Another line of works views constrained policy improvement as Expectation-Maximization algorithm where minimizing KL-term corresponds to the Expectation step, which can be off-policy (Abdolmaleki et al., 2018) or on-policy (Song et al., 2019). From mirror descent perspective, several works also use KL divergence to regularize policy updates (Yang et al., 2019; Tomar et al., 2020; Shani et al., 2020). A recent analysis also points out the advantages of using KL term as a regularizer over a hard constraint (Lazić et al., 2021). Some other works improve the standard trust-region with adaptive clip range (Wang et al., 2019) or off-policy data (Fakoor et al., 2020). Our approach shares similarities with them where we also jointly optimize the approximate expected advantage and KL constraint terms for multiple epochs of minibatch updates. However, we propose a novel dynamic virtual policy to construct the second trust region as a supplement to the traditional trust region defined by the old or previous policy.

## 6 DISCUSSION

We have presented Memory-Constrained Policy Optimization, a new method to regularize each policy update with two-trust regions with respect to one single old policy and another virtual policy representing history of past policies. The new policy is encouraged to stay closer to the region surrounding the policy that performs better. The virtual policy is determined online through a learned attention to a memory of past policies. MCPO is applicable in various problems and settings, less sensitive to hyperparameter changes, and showing better performance in many environments.

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

APPENDIX

## A  METHOD DETAILS

### A.1  THE ATTENTION NETWORK

The attention network is implemented as a feedforward neural network with one hidden layer:

- Input layer: 12 units
- Hidden layer: $N$ units coupled with a dropout layer $p = 0.5$
- Output layer: $N$ units, $\mathrm{softmax}$ activation function

$N$ is the capacity of policy memory. The 12 features of the input $v_{context}$ is listed in Table 3.

### A.2  THE ADVANTAGE FUNCTION

In this paper, we use GAE (Schulman et al., 2015b) as the advantage function for all models and experiments

$$\hat{A}_t = \frac{1}{N_{actor}} \sum_{i}^{N_{actor}} \sum_{k=0}^{T-t-1} (\gamma\lambda)^k \left( r_{t+k}^i + \gamma V \left( s_{t+k+1}^i \right) - V \left( s_{t+k}^i \right) \right)$$

where $\gamma$ is the discounted factor and $N_{actor}$ is the number of actors. The term $r_{t+k}^i + \gamma V \left( s_{t+k+1}^i \right)$ is also known as $V_{target}$ in computing the value loss. Note that Algo. 1 illustrates the procedure for 1 actor. In practice, we use $N_{actor}$ depending on the tasks.

### A.3  THE OBJECTIVE FUNCTION

Following Schulman et al. (2017), our objective function also includes value loss and entropy terms. This is applied to all of the baselines. For example, the complete objective function for MCPO reads

$$L = L^{MCPO} - c_1 \hat{\mathbb{E}}_t \left( V_\theta \left( s_t \right) - V_{target} \left( s_t \right) \right)^2 + c_2 \hat{\mathbb{E}}_t \left[ -\log \left( \pi_{\psi_{old}} \left( \cdot | s_t \right) \right) \right]$$

where $c_1$ and $c_2$ are value and entropy coefficient hyperparameters, respectively. $V_\theta$ is the value network, also parameterized with $\theta$.

## B  EXPERIMENTAL DETAILS

### B.1  BASELINES AND TASKS

All baselines in this paper share the same setting of policy and value networks. Except for TRPO, all other baselines use minibatch training. The only difference is the objective function, which revolves around KL and advantage terms. We train all models with Adam optimizer. We summarize the policy and value network architecture in Table 4.

The baselines ACKTR, PPO[1], TRPO[2] use available public code. They are Pytorch reimplementation of OpenAI's stable baselines, which can reproduce the original performance relatively well. For MDPO, we refer to the authors' source code[3] to reimplement the method. For VMPO, we refer to this open source code[4] to reimplement the method. We implement KL Fixed and KL Adaptive, using objective function defined in Sec. 2.

We use environments from Open AI gyms [5], which are public and using The MIT License. Mujoco environments use Mujoco software[6] (our license is academic lab). Table 5 lists all the environments.

---

[1] https://github.com/ikostrikov/pytorch-a2c-ppo-acktr-gail
[2] https://github.com/ikostrikov/pytorch-trpo
[3] https://github.com/manantomar/Mirror-Descent-Policy-Optimization
[4] https://github.com/YYCAAA/V-MPO_Lunarlander
[5] https://gym.openai.com/envs/
[6] https://www.roboti.us/license.html

| Dimension | Feature | Meaning |
|---|---|---|
| 1 | $d\left(\theta, \psi_{old}\right)$ | "Distance" between $\theta$ and $\psi_{old}$ |
| 2 | $d\left(\theta_{old}, \psi_{old}\right)$ | "Distance" between $\theta_{old}$ and $\psi_{old}$ |
| 3 | $d\left(\theta_{old}, \theta\right)$ | "Distance" between $\theta_{old}$ and $\theta$ |
| 4 | $\hat{\mathbb{E}}_t\left[R_t\left(\psi_{old}\right)\right]$ | Approximate expected advantage of $\psi_{old}$ |
| 5 | $\hat{\mathbb{E}}_t\left[R_t\left(\theta_{old}\right)\right]$ | Approximate expected advantage of $\theta_{old}$ |
| 6 | $\hat{\mathbb{E}}_t\left[R_t\left(\theta\right)\right]$ | Approximate expected advantage of $\theta$ |
| 7 | $\hat{\mathbb{E}}_t\left[-\log\left(\pi_{\psi_{old}}\left(\cdot|s_t\right)\right)\right]$ | Approximate entropy of $\psi_{old}$ |
| 8 | $\hat{\mathbb{E}}_t\left[-\log\left(\pi_{\theta_{old}}\left(\cdot|s_t\right)\right)\right]$ | Approximate entropy of $\theta_{old}$ |
| 9 | $\hat{\mathbb{E}}_t\left[-\log\left(\pi_{\theta}\left(\cdot|s_t\right)\right)\right]$ | Approximate entropy of $\theta$ |
| 10 | $\hat{\mathbb{E}}_t\left(V_{\psi_{old}}\left(s_t\right) - V_{target}\left(s_t\right)\right)^2$ | Value loss of $\psi_{old}$ |
| 11 | $\hat{\mathbb{E}}_t\left(V_{\theta_{old}}\left(s_t\right) - V_{target}\left(s_t\right)\right)^2$ | Value loss of $\theta_{old}$ |
| 12 | $\hat{\mathbb{E}}_t\left(V_{\theta}\left(s_t\right) - V_{target}\left(s_t\right)\right)^2$ | Value loss of $\theta$ |

Table 3: Features of the context vector.

| Input type | Policy/Value networks |
|---|---|
| Vector | 2-layer feedforward net (tanh, h=64) |
| Image | 3-layer ReLU CNN with kernels $\{32/8/4, 64/4/2, 32/3/1\}$+2-layer feedforward net (ReLU, h=512) |

Table 4: Network architecture shared across baselines.

| Tasks | Continuous action | Gym category |
|---|---|---|
| Pendulum-v0 | X | Classical |
| LunarLander-v2 | | Box2d |
| BipedalWalker-v3 | ✓ | |
| Unlock-v0 | X | MiniGrid |
| UnlockPickup-v0 | | |
| MuJoCo tasks (v2): HalfCheetah Walker2d, Hopper, Ant Humanoid, HumanoidStandup | ✓ | MuJoCo |
| Atari games (NoFramskip-v4): Beamrider, Breakout Enduro, Gopher Seaquest, SpaceInvaders | X | Atari |
| BipedalWalkerHardcore-v3 | ✓ | Box2d |

Table 5: Tasks used in the paper.

| Hyperparameter | Pendulum | LunarLander | BipedalWalker | MiniGrid | BipedalWaker Hardcore |
|---|---|---|---|---|---|
| Horizon $T$ | 2048 | 2048 | 2048 | 2048 | 2048 |
| Adam step size | $3 \times 10^{-4}$ | $3 \times 10^{-4}$ | $3 \times 10^{-4}$ | $3 \times 10^{-4}$ | $3 \times 10^{-4}$ |
| Num. epochs $K$ | 10 | 10 | 10 | 10 | 10 |
| Minibatch size $B$ | 64 | 64 | 64 | 64 | 64 |
| Discount $\gamma$ | 0.99 | 0.99 | 0.99 | 0.99 | 0.99 |
| GAE $\lambda$ | 0.95 | 0.95 | 0.95 | 0.95 | 0.95 |
| Num. actors $N_{actor}$ | 4 | 4 | 32 | 4 | 128 |
| Value coefficient $c_1$ | 0.5 | 0.5 | 0.5 | 0.5 | 0.5 |
| Entropy coefficient $c_2$ | 0 | 0 | 0 | 0 | 0 |

Table 6: Network architecture shared across baselines on Pendulum, LunarLander, BipedalWalker, MiniGrid and BipedalWaker Hardcore

| Model | Speed (env. steps/s) |
|---|---|
| MCPO (N=5) | 1,170 |
| MCPO (N=10) | 927 |
| MCPO (N=40) | 560 |
| PPO | 1,250 |

Table 7: Computing cost of MCPO and PPO on Pendulum.

## B.2 DETAILS ON CLASSICAL CONTROL

For these tasks, all models share hyperparameters listed in Table 6. Besides, each method has its own set of additional hyperparameters. For example, PPO, KL Fixed and KL Adaptive have $\epsilon$, $\beta$ and $d_{targ}$, respectively. These hyperparameters directly control the conservativeness of the policy update for each method. For MDPO, $\beta$ is automatically reduced overtime through an annealing process from 1 to 0 and thus should not be considered as a hyperparameter. However, we can still control the conservativeness if $\beta$ is annealed from a different value $\beta_0$ rather 1. We realize that tuning $\beta_0$ helped MDPO (Table 1). We quickly tried with several values $\beta_0$ ranging from 0.01 to 10 on Pendulum, and realize that only $\beta_0 \in \{0.5, 1, 2\}$ gave reasonable results. Thus, we only tuned MDPO with these $\beta_0$ in other tasks. For VMPO there are many other hyperparameters such as $\eta_0$, $\alpha_0$, $\epsilon_\eta$ and $\epsilon_\alpha$. Due to limited compute, we do not tune all of them. Rather, we only tune $\alpha_0$-the initial value of the Lagrange multiplier that scale the KL term in the objective function. We refer to the paper's and the code repository's default values of $\alpha_0$ to determine possible values $\alpha_0 \in \{0.1, 1, 5\}$. For our MCPO, we can tune several hyperparameters such as $N$, $\beta_{min}$, and $\beta_{max}$. However, for simplicity, we only tune $N \in \{5, 10, 40\}$ and fix $\beta_{min} = 0.01$ and $\beta_{max} = 10$.

On our machines using 1 GPU Tesla V100-SXM2, we measure the running time of MCPO with different $N$ compared to PPO on Pendulum task, which is reported in Table 7. As $N$ increases, the running speed of MCPO decreases. For this reason, we do not test with $N > 40$. However, we realize that with $N = 5$ or $N = 10$, MCPO only runs slightly slower than PPO. We also realize that the speed gap is even reduced when we increase the number of actors $N_{actor}$ as in other experiments. In terms of memory usage, maintaining a policy memory will definitely cost more. However, as our policy, value and attention networks are very simple. The maximum storage even for $N = 40$ is less than 5GB.

In addition to the configurations reported in Table 1, for KL Fixed and PPO, we also tested with extreme values $\beta = 10$ and $\epsilon \in \{0.5, 0.8\}$. Figs. 6, 7 and 8 visualize the learning curves of all configurations for all models.

## B.3 DETAILS ON MINIGRID NAVIGATION

Based on the results from the above tasks, we pick the best signature hyperparameters for the models to use in this task as in Table 8. In particular, for each model, we rank the hyperparameters per task (higher rank is better), and choose the one that has the maximum total rank. For hyperparameters

| Model | Chosen hyperparameter |
|---|---|
| KL Adaptive | $d_{targ} = 0.01$ |
| KL Fixed | $\beta = 0.1$ |
| PPO | $\epsilon = 0.2$ |
| MDPO | $\beta_0 = 0.5$ |
| VMPO | $\alpha_0 = 1$ |
| MCPO | $N = 10$ |

Table 8: Signature hyperparameters used in MiniGrid tasks.

| Hyperparameter | Mujoco | Atari |
|---|---|---|
| Horizon $T$ | 2048 | 128 |
| Adam step size | $3 \times 10^{-4}$ | $2.5 \times 10^{-4}$ |
| Num. epochs $K$ | 10 | 4 |
| Minibatch size $B$ | 32 | 32 |
| Discount $\gamma$ | 0.99 | 0.99 |
| GAE $\lambda$ | 0.95 | 0.95 |
| Num. actors $N_{actor}$ | 16 | 32 |
| Value coefficient $c_1$ | 0.5 | 1.0 |
| Entropy coefficient $c_2$ | 0 | 0.01 |

Table 9: Network architecture shared across baselines on Mujoco and Atari

that share the same total rank, we prefer the middle value. The other hyperparameters for this task is listed in Table 6.

### B.4 DETAILS ON MUJOCO

For shared hyperparameters, we use the values suggested in the PPO's paper, except for the number of actors, which we increase to 16 for faster training as our models are trained for 10M environment steps (see Table 9).

For the signature hyperparameter of each method, we select some of the reasonable values. For PPO, the authors already examined with $\epsilon \in \{0.1, 0.2, 0.3\}$ on the same task and found 0.2 the best. This is somehow backed up in our previous experiments where we did not see major difference in performance between these values. Hence, seeking for other $\epsilon$ rather than the optimal $\epsilon = 0.2$, we ran our PPO implementation with $\epsilon \in \{0.2, 0.5, 0.8\}$. For TRPO, the authors only used the KL radius threshold $\delta = 0.01$, which may be already the optimal hyperparameter. Hence, we only tried $\delta \in \{0.005, 0.01\}$. The results showed that $\delta = 0.005$ always performed worse. For MCPO and Mean $\psi$, we only ran with extreme $N \in \{5, 40\}$. For MDPO, we still tested with $\beta_0 \in \{0.5, 1, 2\}$. Full learning curves with different hyperparameter are reported in Fig. 9. Learning curves including TRGPPO[7] are reported in Fig. 10

### B.5 DETAILS ON ATARI

For shared hyperparameters, we use the values suggested in the PPO's paper, except for the number of actors, which we increase to 32 for faster training (see Table 9). For the signature hyperparameter of the baselines, we used the recommended value in the original papers. For MCPO, we use $N = 10$ to balance between running time and performance. Table 10 shows the values of these hyperparameters.

We also report the average normalized human score (mean and median) of the models over 6 games in Table 11. As seen, MCPO is significantly better than other baselines in terms of both mean and median normalized human score. We also report full learning curves of models and normalized human score including TRGPPO in 9 games in Fig. 4 and Table 12, respectively.

---

[7]We use the authors' source code `https://github.com/wangyuhuix/TRGPPO` using default configuration. Training setting is adjusted to follow the common setting as for other baselines (see Table 9).

| Model | Chosen hyperparameter |
|-------|----------------------|
| PPO | $\epsilon = 0.2$ |
| ACKTR | $\delta = 0.01$ |
| VMPO | $\alpha_0 = 5$ |
| MCPO | $N = 10$ |

Table 10: Signature hyperparameters used in Atari tasks.

| Model | Mean | Median |
|-------|------|--------|
| PPO | 154.63 | 48.36 |
| ACKTR | 266.73 | 21.99 |
| VMPO | 412.19 | 20.85 |
| MCPO | **300.25** | **100.28** |

Table 11: Average normalized human score over 6 games. For each model, the performance of each run is measured by the best checkpoint during training over 10 million frames, then take average over 5 runs.

To verify whether MCPO can maintain its performance over longer training, we examine Atari training for 40 million frames. As shown in Fig. 5, MCPO is still the best performer in this training regime.

## B.6 DETAILS ON ABLATION STUDY

In this section, we give more details on the ablated baselines.

- **No $\psi$** We only changed the objective to

$$
\begin{aligned}
L_1(\theta) = \hat{\mathbb{E}}_t \left[ rat_t(\theta) \hat{A}_t \right] \\
- \beta \hat{\mathbb{E}}_t \left[ KL \left[ \pi_{\theta_{old}}(\cdot | s_t), \pi_\theta(\cdot | s_t) \right] \right]
\end{aligned}
\tag{6}
$$

  where $\beta$ is still determined by the $\beta$-switching rule.

- **Annealed $\beta$** We determine the $\beta$ in Eq. 1 by MDPO's annealing rule, a.k.a, $\beta_i = 1.0 - \frac{i}{T_{total}}$ where $T_{total}$ is the total number of training policy update steps and $i$ is the current update step. We did not test with other rules such as fixed or adaptive $\beta$ as we realize that MDPO is often better than KL Fixed and KL Adaptive in our experiments, indicating that the annealed $\beta$ is a stronger baseline.

- **Frequent writing** We add a new policy to $\mathcal{M}$ at every policy update step.

- **Uniform writing** Inspired by the uniform writing mechanism in Memory-Augmented Neural Networks (Le et al., 2019), we add a new policy to $\mathcal{M}$ at every interval of 10 update steps. The interval size could be tuned to get better results but it would require additional effort, so we preferred our conditional writing over this one.

- **Mean $\psi$** The virtual policy is determined as

| Model | Mean | Median |
|-------|------|--------|
| PPO | 131.19 | 52.85 |
| ACKTR | 195.52 | 25.30 |
| VMPO | 18.20 | 13.56 |
| TRGPPO | 116.80 | 43.24 |
| MCPO | **229.99** | **65.78** |

Table 12: Average normalized human score over 9 games. For each model, the performance of each run is measured by the best checkpoint during training over 10 million frames, then take average over 5 runs.

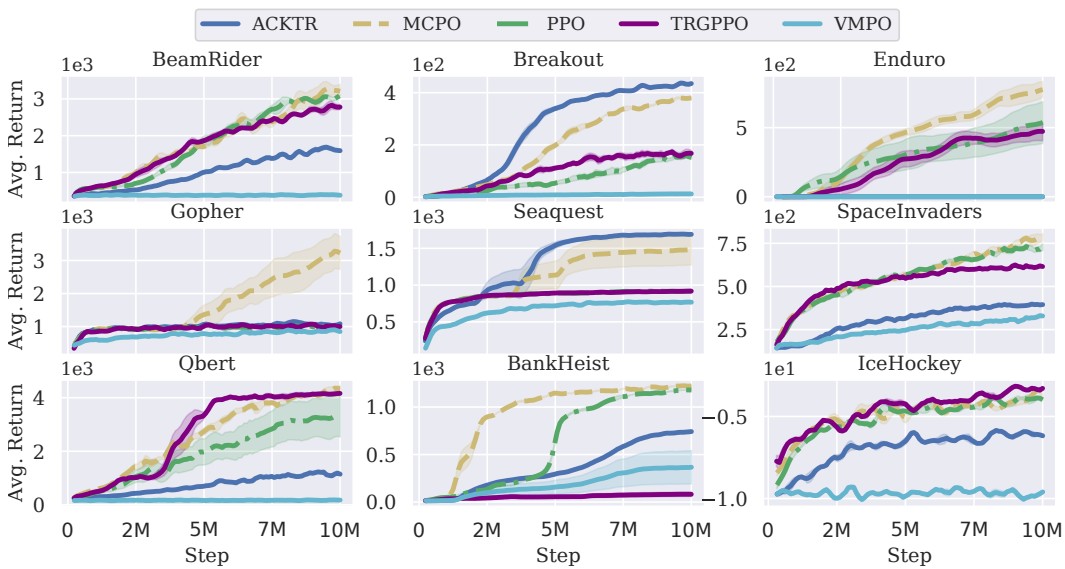

Figure 4: Atari games: learning curves (mean and std. over 5 runs) across 10M training steps.

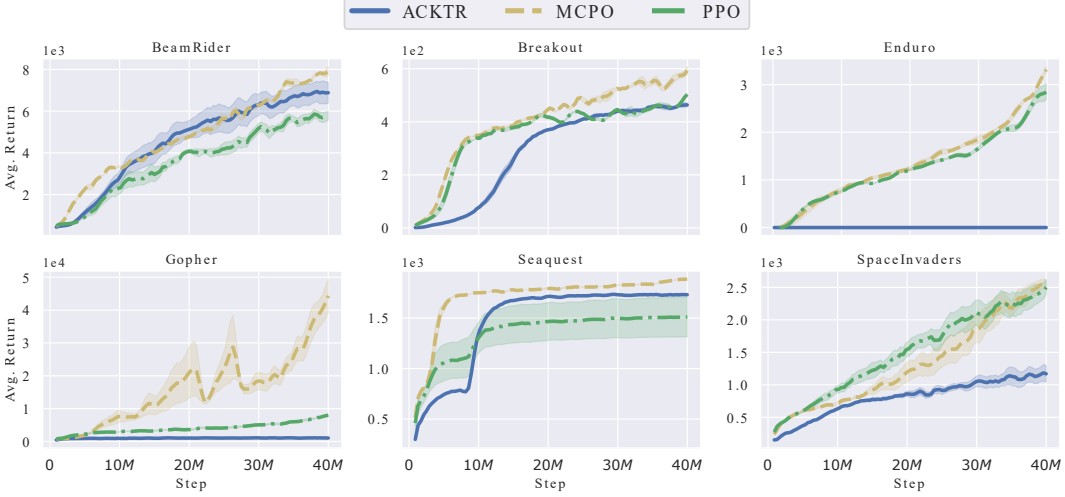

Figure 5: Atari games: learning curves (mean and std. over 5 runs) across 40M training steps.

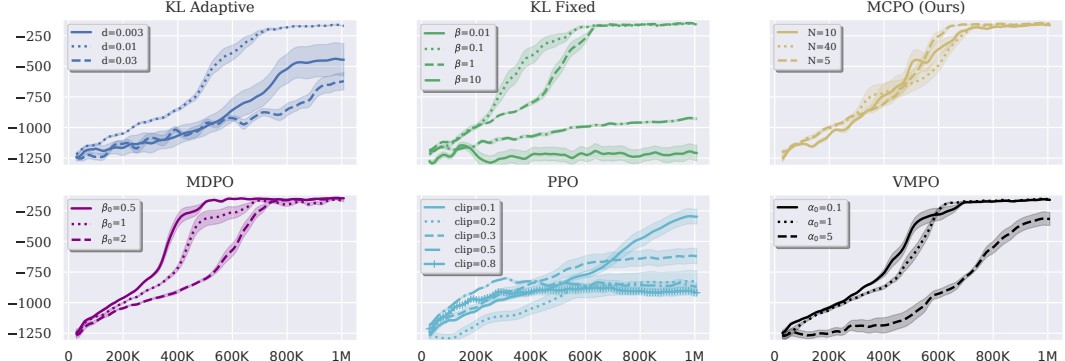

Figure 6: Pendulum-v0: learning curves (mean and std. over 5 runs) across 1M training steps.

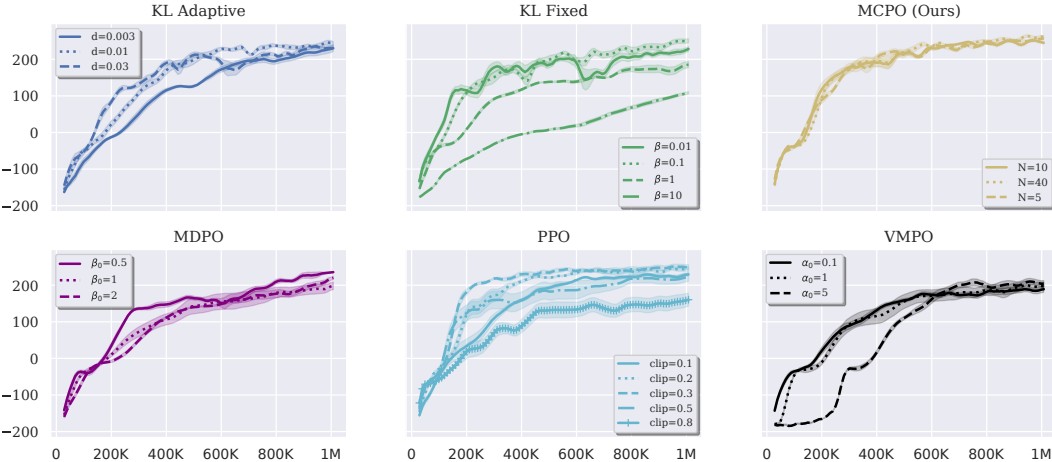

Figure 7: LunarLander-v2: learning curves (mean and std. over 5 runs) across 5M training steps.

$$\psi = \sum_j^{|\mathcal{M}|} \theta_j \tag{7}$$

- **Half feature** We only use features from 1 to 6 listed in Table 3.

The other baselines including KL Adaptive, KL Fixed, MDPO, PPO, and VMPO are the same as in B.2. The full learning curves of all models with different hyperparameters are plotted in Fig. 11.

## C   THEORETICAL ANALYSIS OF MCPO

In this section, we explain the design of our objective function $L_1$ and $L_2$. Similar to Schulman et al. (2015a), we can construct a theoretically guaranteed version of our practical objective functions that ensures monotonic policy improvement.

First, we explain the design of $L_1$ by recasting $L_1$ as

$$
\begin{aligned}
L_{1\theta_{old}}(\theta) = L_{\theta_{old}}(\theta) \\
- C_1 D_{KL}^{max}(\theta_{old}, \theta) \\
- C_2 D_{KL}^{max}(\psi, \theta)
\end{aligned}
$$

where $L_{\theta_{old}}(\theta) = \eta(\pi_{\theta_{old}}) + \sum_s \rho_{\pi_{\theta_{old}}}(s) \sum_a \pi_\theta(a|s) A_{\pi_{\theta_{old}}}(s,a)$–the local approximation to the expected discounted reward $\eta(\theta)$, $D_{KL}^{max}(a,b) = \max_s KL(\pi_a(\cdot|s), \pi_b(\cdot|s))$, $C_1 = \frac{4 \max_{s,a} |A_\pi(s,a)| \gamma}{(1-\gamma)^2}$ and $C_2 > 0$.

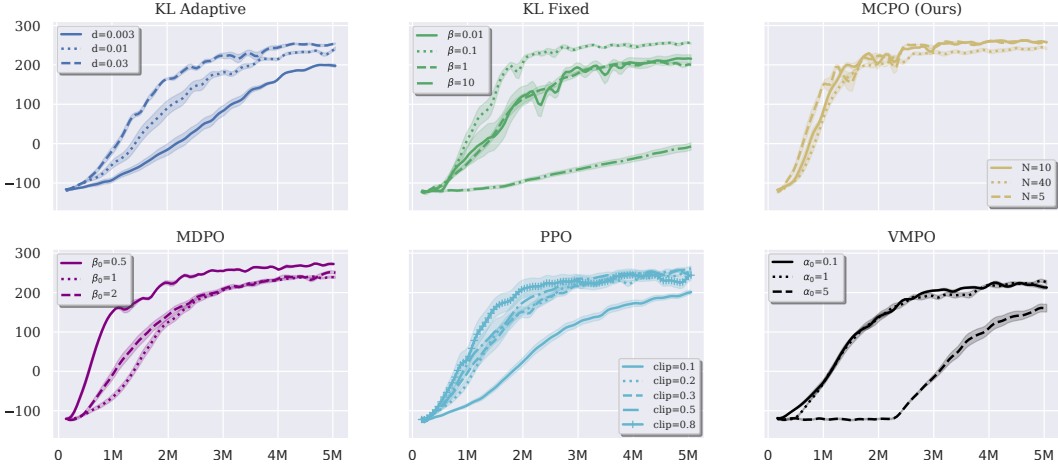

Figure 8: BipedalWalker-v3: learning curves (mean and std. over 5 runs) across 1M training steps.

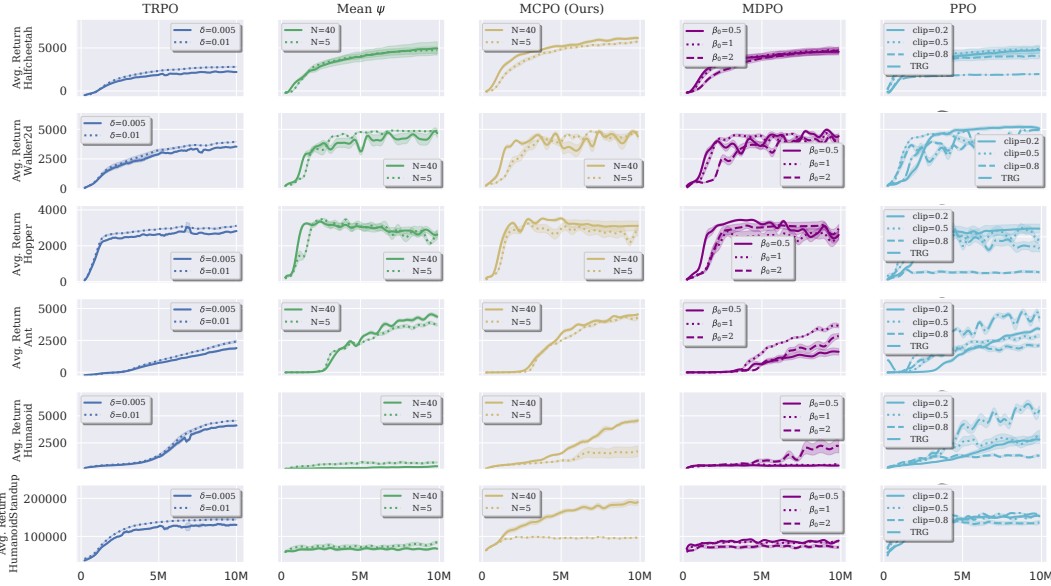

Figure 9: Mujoco: learning curves (mean and std. over 5 runs) across 10M training steps.

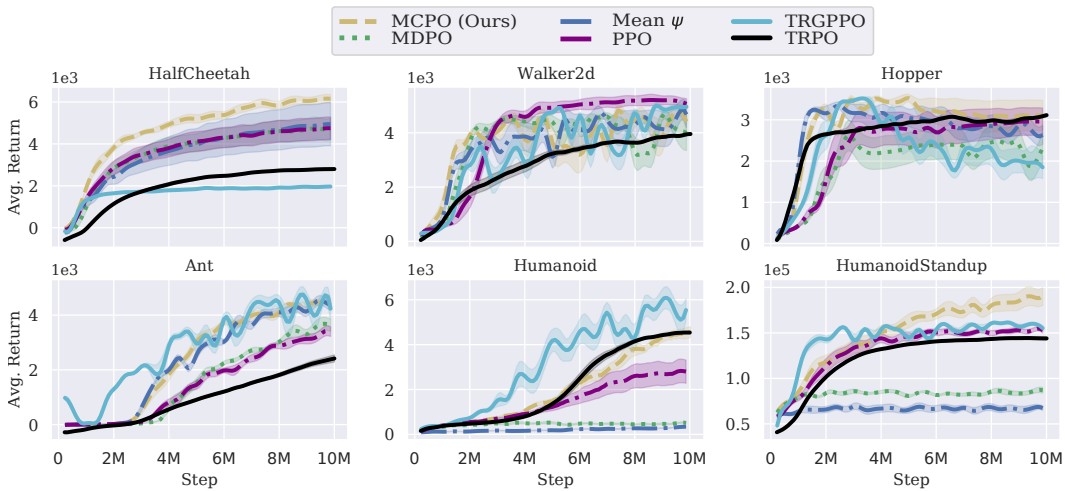

Figure 10: Mujoco: learning curves (mean and std. over 5 runs) across 10M training steps.

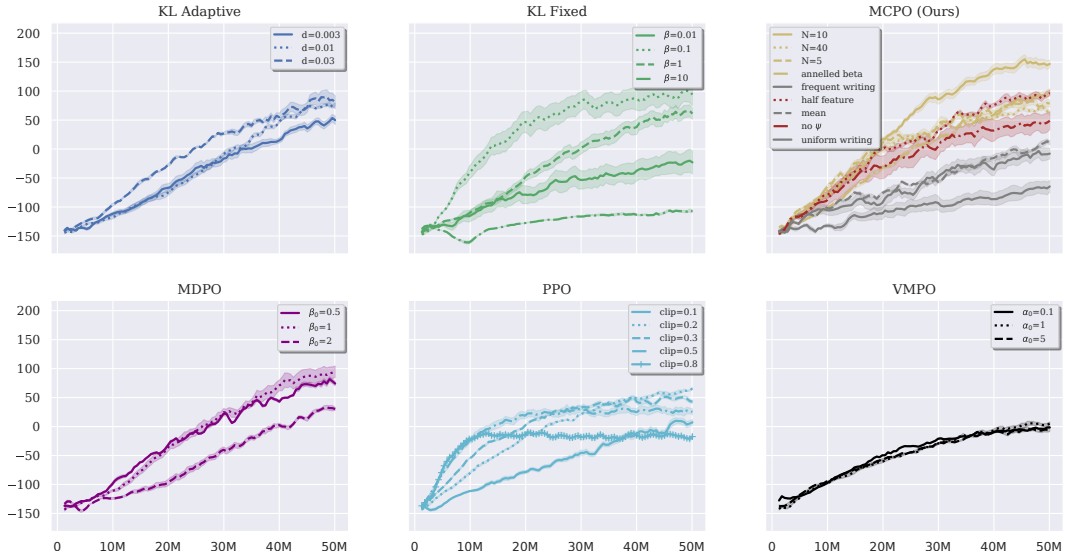

Figure 11: BibedalWalkerHardcore-v3: learning curves (mean and std. over 5 runs) across 50M training steps.

As the KL is non-negative, $L_{1\theta_{old}}(\theta) \leq L_{\theta_{old}}(\theta) - C_1 D_{KL}^{max}(\theta_{old}, \theta)$. According to Schulman et al. (2015a), the RHS is a lower bound on $\eta(\theta)$, so $L_1$ is also a lower bound on $\eta(\theta)$ and thus, it is reasonable to maximize the practical $L_1$, which is an approximation of $L_{1\theta_{old}}$.

Next, we show that by optimizing both $L_1$ and $L_2$, we can interpret our algorithm as a monotonic policy improvement procedure. As such, we need to reformulate $L_2$ as

$$L_{2\theta_{old}}(\psi) = L_{\theta_{old}}(\psi) - C_1 D_{KL}^{max}(\theta_{old}, \psi)$$

Note that compared to the practical $L_2$ (as defined in the main paper on page 5), we have introduced here an additional $KL$ term, which means we need to find $\psi$ that is close to $\theta_{old}$ and maximizes the approximate advantage $L_{\theta_{old}}(\psi)$. As we maximize $L_{2\theta_{old}}(\psi)$, the maximizer $\psi$ satisfies

$$L_{2\theta_{old}}(\psi) \geq L_{2\theta_{old}}(\theta_{old}) = L_{\theta_{old}}(\theta_{old})$$

We also have

$$\eta(\theta) \geq L_{1\theta_{old}}(\theta) \tag{8}$$
$$\eta(\theta_{old}) = L_{\theta_{old}}(\theta_{old}) \leq L_{2\theta_{old}}(\psi)$$
$$= L_{\theta_{old}}(\psi) - C_1 D_{KL}^{max}(\theta_{old}, \psi)$$
$$= L_{1\theta_{old}}(\psi) \tag{9}$$

Subtracting both sides of Eq. 9 from Eq. 8 yields

$$\eta(\theta) - \eta(\theta_{old}) \geq L_{1\theta_{old}}(\theta) - L_{1\theta_{old}}(\psi)$$

Thus by maximizing $L_{1\theta_{old}}(\theta)$, we guarantee that the true objective $\eta(\theta)$ is non-decreasing.

Although the theory suggests that the practical $L_2$ could be $L_2^* = \hat{\mathbb{E}}_t[R_t(\psi(\varphi)) - C_1 KL[\pi_{\theta_{old}}(\cdot|s_t), \pi_\psi(\cdot|s_t)]]$, it would require additional tuning of $C_1$. More importantly, optimizing an objective in form of $L_2^*$ needs a very small step size, and could converge slowly. Hence, we simply discard the KL term and only optimize $L_2 = \hat{\mathbb{E}}_t[R_t(\psi(\varphi))]$ instead. Empirical results show that using this simplification, MCPO's learning curves still generally improve monotonically over training time.

