# OpenReview forum: "Memory-Constrained Policy Optimization"
_ICLR.cc/2022/Conference — ICLR 2022 Submitted_

### Official Review · Reviewer_HBno · 2021-11-02

**Correctness:** 2
**Technical Novelty And Significance:** 2
**Empirical Novelty And Significance:** 2
**Recommendation:** 3
**Confidence:** 3

**Main Review:**

I have several concerns with this paper. In its current form, the paper is somewhat confusing to the reader, and could use from a clearer rewrite. More importantly, the novelty of the paper appears to be minor and the the design decisions of the author's algorithm do not seem to be very well explained.

- Why is there a need to have two trust regions? Following the author's argument for their  virtual policy, would it not be sufficient to use simply a properly designed trust region with respect to to it?

- The design of the virtual policy seems to be completely ad-hoc and unmotivated. Why are the policies in the memory buffer mixed by a weighted average of their parameters? A big reason to use the KL divergence is due to the fact that similarity in the parameter space may not lead to similarity in policy output (specially for parametrization such as neural networks).

**Summary Of The Paper:**

In this work, the authors propose the use a trust-region method in the vein of Proximal Policy Optimization (PPO). In contrast to previous work, the author propose to use two trust regions instead of a single one. Instead of constraining the policy to stay near to the previous policy, the author's proposed algorithm includes an extra virtual policy constructed from a memory buffer of past policies.

**Summary Of The Review:**

In its current form, I cannot recommend this paper for publication.

---

> ### Author Response · Authors · 2021-11-18
> **Response to Reviewer HBno**
>
> **"Why is there a need to have two trust regions?"**
>
> We need to include the old policy's trust region because, in theory, constraining policy updates using the old policy's trust region guarantees monotonic improvement (see TRPO paper). However, in practice, the old policy can be suboptimal and may not induce monotonic improvement. This motivates us to employ an additional trust region to regulate the update in case the old policy's trust region is bad. By doing so, we hope that we can maintain the theoretical property of trust-region update while enabling a more robust optimization that works well even when the ideal setting for theoretical assurance does not hold. We proved that our objective of two trust regions as in Eq. 1, could also ensure monotonic improvement (see Appendix C). Simply using one properly designed trust region (different from the old policy's) cannot ensure that property.
>
> **"The design of virtual policy ..."**
>
> We note that we do *not* use a weighted average over policies' parameters to enforce policy similarity. Rather, we construct a policy using a mixture of parameters to facilitate *policy attention*. We can view it as an efficient process of selecting the best policy in the policy buffer $\mathcal{M}$ (see more discussion regarding this point in subsection "Objective function" on page 5). This policy (virtual policy) is used to constrain the new policy via KL divergence. Hence, it encourages the new policy's outputs to be close to the virtual policy's outputs as expected.

---

### Official Review · Reviewer_RkGG · 2021-11-02

**Correctness:** 4
**Technical Novelty And Significance:** 3
**Empirical Novelty And Significance:** 2
**Recommendation:** 5
**Confidence:** 4

**Main Review:**

Pros: The overall paper is easy to follow, and the motivation is also attractive. The suboptimality of the previous policy in the trust region indeed cases undesirable constraints for follow-up optimization. Experimental results also validate the effectiveness algorithm.

Cons: 1. The major concern I have is how MCPO addresses such suboptimality problems. Use the example in the paper, when the previous policy is at the local optimum, the new policy will also be constrained around the local optimum. But how MCPO policy can escape from this optimum? There might be some history policies that explore other regions of the landscape, how are these policies guaranteed to be chosen as the additional constraint? Otherwise, adding the worse history policies which lie in the convex hull of the local optimum will even harm the policy optimization.\
2. It would be helpful to provide some theorem support, maybe under ideal cases (e.g., when the memory is not constrained).\
3. In experiments Fig. 2 and 3, the curves are not converging. Could you run for more iterations?


**Summary Of The Paper:**

The paper concerns the suboptimality of the "previous" policy in the trust-region policy optimization algorithm. And the authors propose MemoryConstrained Policy Optimization (MCPO) that uses an additional trust-region relying on the virtual policy, the one that represents the history of policies.

**Summary Of The Review:**

The motivation of the suboptimality is sound, but the algorithm to solve this problem is not very convincing.

---

> ### Author Response · Authors · 2021-11-18
> **Response to Reviewer RkGG**
>
> **"... how MCPO addresses such suboptimality ..."**
>
> By optimizing $L_2$ in Eq. (5), we are searching for the best virtual policy that lies in the convex hull of policies in $\mathcal{M}$. As $\mathcal{M}$ also stores the old policy, the found virtual policy should be at least equal or better than the old policy. Fig. 2 b empirically confirms this point. The role of optimizing $L_2$ is also validated in Sec. 4.3 and 4.5, where we compare optimized virtual policy (Learned $\psi$) with average virtual policy (Mean $\psi$). In many cases, Learned $\psi$ outperforms other baselines significantly whereas Mean $\psi$ cannot, indicating that optimizing $L_2$ enables choosing a good virtual policy.
>
> **"... theorem support ..."**
>
> Due to space constraints, we presented a theoretical analysis on the monotonic improvement of MCPO in Appendix C. We proved that by designing $L_1$ as in Eq. 1 and optimizing $L_2$ successfully, MCPO also guarantees monotonic improvement similarly as TRPO does.
>
> **"... Could you run for more iterations?"**
>
> Fig. 2 b, we only saved and visualized several checkpoints of $\theta_{old}$ and $\psi$ across update steps (there are about 8000 update steps in total). Fig. 2 b  plotted the performance of the last checkpoint we saved. Since the learning rate is annealed to 0 following PPO's setting,  eventually, both the virtual and old policy will converge to the current policy, and hence share the same performance at the end of training.
>
> Fig. 3, we chose the setting of training for 10 million frames, which is common to measure sample-efficiency of RL methods in Atari benchmark. Also, it was suitable for our compute budget. For Atari games, the results will often be better with more training. To show that training for more iterations is not an issue to MCPO, we will train models for 40 million frames. So far, we have finished and added training curves for the first 3 games and the most 3 competitive baselines (learning curves in Appendix Fig. 5 in this revision). The results show that MCPO still outperforms other baselines after 40M frames. In the next revision, we will provide the remaining training curves for this setting.

---

### Official Review · Reviewer_JAMC · 2021-11-04

**Correctness:** 4
**Technical Novelty And Significance:** 3
**Empirical Novelty And Significance:** 3
**Recommendation:** 8
**Confidence:** 4

**Main Review:**

I think this is quite a well-written paper. While other works have explored the idea of enforcing trust region on all or several past policies, the use of trust region constraints along with an attention mechanism for past policies is to the best of my knowledge novel. My detailed comments are as follows:

- The presentation of the paper is mostly clear and overall I found it quite easy to read.
- The experiments are very well-run, and I think it is one of the main strength of this paper. The authors demonstrated results on a large set of environments. The ablation studies are most appreciated and included discussions on most of the key components of the algorithm.
- Can you comment briefly on the intuition behind conditional writing?
- How did you determine the context vector features? You also mentioned in your ablation studies that using fewer context features causes a loss of information. Did you find any features that were particularly important?
- (Minor) I suggest the authors change the heading of Section 2, Constrained Policy Optimization (CPO) (Achiam et al. 2017) is a well known paper for constrained RL and the heading may cause some confusion.
- (Minor typo) Last paragraph of page 3: ...dynamically 'weigh' the whole KL....

**Summary Of The Paper:**

This paper proposes a new trust-region based policy optimization method with two trust regions constraints for policy updates which enforces the proximity of the current policy to (1) the old policy prior to the current update and (2) a virtual policy constructed using an attention mechanism and memory buffer which stores several past policies. In addition a new algorithm based the the aforementioned ideas + minibatch updates is also proposed, which is shown to perform well on a series of high-dimensional control tasks.

**Summary Of The Review:**

Overall, I think this paper proposed some very novel and insightful ideas with well-run experiments that would be beneficial to the community. I think this paper passes the bar for acceptance.

---

> ### Author Response · Authors · 2021-11-18
> **Response to Reviewer JAMC**
>
> **"... intuition behind conditional writing ..."**
>
> We want to construct an informative memory buffer  $\mathcal{M}$, whose size is limited. If we add a policy to  $\mathcal{M}$ at any update step unconditionally,  $\mathcal{M}$ is likely to be filled with similar policies and fails to represent a diverse policy population (and hence, lower the chance of finding the optimal virtual policy). Our conditional writing is an efficient way to add policies that are distinct enough. Using the distance between the current virtual policy ($\psi$-representing  $\mathcal{M}$) and the old policy $\theta_{old}$  as a soft threshold, we only add a policy $\theta$ to  $\mathcal{M}$ if its distance from $\psi$ surpasses the threshold. Intuitively, $\theta$ must be further from $\psi$ than $\theta_{old}$ to be added to  $\mathcal{M}$. Our ablation study confirms the effectiveness of this intuition (see 4.5).
>
> **"... context vector features ..."**
>
> For simplicity, the context only involves $\theta$, $\theta_{old}$ and the previous $\psi_{old}$. From these three policies, we tried to extract all possible information. The information should be cheap to extract and dependent on the current data, so we prefer features extracted from the outputs of these policies (value, entropy, distance, return, etc.). We have not conducted a contribution analysis. Yet, intuitively, the most important features should be the empirical returns, values associated with each policy and the distances, which gives a good hint of which virtual policy will yield high performance (e.g., a virtual policy that is closer to the policy that obtained high return and low value loss).
>
> **Minor comments**
>
> Thank you for pointing out these problems. We have revised accordingly.

---

### Official Review · Reviewer_9u6G · 2021-11-06

**Correctness:** 4
**Technical Novelty And Significance:** 2
**Empirical Novelty And Significance:** 2
**Recommendation:** 5
**Confidence:** 5

**Main Review:**

I agree with the authors that enforcing policy updates in constrained on-policy methods considering only previous policy can be suboptimal and it is better to utilize the history of past policies to make the update more efficient and it can be a right direction to improve the performance of on-policy methods. That being said, there are some concerns in terms of novelty, related works, and experimental results:

- While using attention weights to combine the past policies is interesting and shows some promising results and I consider this is the main novelty of this paper, the idea of improving trust region constraint in on-policy methods has been extensively studied in recent years ( which is not a weakness though). However, one of the main concerns with this paper is it fails to discuss and compares with two important related works [1,2] which are more relevant to this paper. Both [1,2] proposed ways to improve constrained on-policy methods by introducing new ways to impose trust regions constraint: [1] propose a way to adaptively adjusts the clipping range within the trust region and [2] introduced a way to adaptively enforce soft trust region between current policy and all previous ones. Comparing and discussing these works in this paper, can help to highlight the contribution of this work and to better situate this work against them.

- Although this paper used various benchmarks to compare with their work, I found the experimental results are not conclusive. Specifically, it is not clear which method is the best performing method on Atari and major experiments ( e.g. this method only shows better results Gopher and Enduro not others). Another concern is that the choice of Atari games seems very arbitrary. What is the logic to select only these games? Does your method only show good performance on these environments, not the rest?  In addition, it is really hard to come to any conclusion just by looking at Table 2 without seeing the learning curves for these experiments that summarize these results.  I have to acknowledge that the authors did a good job in regard to the ablation studies. But unfortunately this has not been the case for comparison with other methods. It is better to show clear improvement on two benchmarks rather than 5 different ones without a clear conclusion.

- The paper is written fairly well, except for the experimental section as it doesn't have a smooth flow as other sections. I highly suggest improving presentation in that section.

- It is not clear to me how the choice of $\beta$ affects the two trust regions. Author explained on page 4, but it is confusing. Can the authors explain in detail why Eq2 makes sense? In addition, you said "The intuition is that we encourage the enforcement of the constraint when $\pi_\theta$ is too far from $\theta_{old}$ using the distance between $\theta_{old}$ and $\theta_{\phi}$ as a reference" if that is the case, why shouldn't you use beta ($1-\beta$) for previous policy and ($\beta$) for virtual policy then?


[1] Trust Region-Guided Proximal Policy Optimization
https://arxiv.org/abs/1901.10314

[2] P3O: Policy-on Policy-off Policy Optimization
https://arxiv.org/abs/1905.01756


**Summary Of The Paper:**

This paper proposes a new approach to regularize on-policy methods utilizing two soft trust region terms. The first term encourages the new policy to stay close to old policy like PPO and the second term enforces the new policy to stay close to the combination of past policies called virtual policy. The virtual policy is built by a convex combination of past policies in the memory through attention weights. The motivation to use attention is to ensure the virtual policy has good performance on current data. To evaluate their method, this paper uses 3 control tasks (i.e. Pendulum, LunarLander and Bipedal- Walker), MiniGrid library for sparse reward environments evaluation, 6 Mujoco tasks, and 6 Atari games.

**Summary Of The Review:**

In general, this paper's proposed method has merits. But unfortunately, it missed some important related works and experimental results are not conclusive enough yet.

---

> ### Author Response · Authors · 2021-11-18
> **Response to Reviewer 9u6G**
>
> **"... two important related works ..."**
>
> Thank you for suggesting two relevant related works. We have mentioned them in this revision's Sec. 5 and will add more discussion (if more space is given) as below.
>
> [1] augments PPO-clip with adaptive clip range. We instead advocate using 2 trust regions and apply our idea to a weaker backbone: PPO-penalty.  Our method is also different from [2] as we do not use off-policy data/gradients to update the policy. Our solution is novel as it learns to generate the optimal past policy to restrain the current policy, rather than using all previous policies to enforce trust regions as in [2]. By doing so, we may avoid the problem mentioned in [2]'s Sec. 3.4.
>
> To make the comparison comprehensive, we have added a performance comparison between MCPO and TRGPPO [1] in the Appendix (we will move it to the main manuscript if more space is given). We decided not to include [2] in our experiments as [2] requires a replay buffer and off-policy data, which does not follow our purely on-policy setting. As shown below, compared to [1], our method is significantly better in 2/6 and 6/6 Mujoco and Atari tasks, respectively while only statistically worse in Humanoid.
>
> |Mujoco|HalfCheetah|Walker2d|Hopper|Ant|Humanoid|HumanoidStandup|
> |---|---|---|---|---|---|---|
> |TRGPPO|2811±114|5009±391|3713±275|4796±837|**6242±1192**|162185±3755|
> |MCPO|**6173±595**|5120±588|3620±252|4673±249|4848±711|**19540±32801**|
>
> |Atari|Beamrider|Breakout|Enduro|Gopher|Seaquest|SpaceInvaders|
> |---|---|--|---|--|--|--|
> |TRGPPO|2959±203|200.0±21.6|483.2±129.1|1150.4±40|919±90|652.6±87.5|
> |MCPO|**3468±350**|**388.9±8.6**|**782.5±113.4**|**3460±1060**|**1486±413**|**839.6±34.7**|
>
> Bold denotes statistically better one (Cohen effect size>0.5). See Appendix's Figs. 4 and 10 for learning curves
>
> **"... experimental results are not conclusive ..."**
>
> Our method is often the best performer in Classical Control, Navigation and Mujoco tasks (see Sec. 4.1, 4.2 and 4.3). For Atari games, although our model is only the best in 2/6 games, on average, in terms of normalized human score (NHS), ours is the best (see below). In this revision, we have reported this metric in Appendix Table 11 to make the comparison clearer.
>
> |Model|Mean NHS|Median NHS|
> |----|---|---|
> |PPO|154.63|48.36|
> |ACKTR|266.73|21.99|
> |VMPO|15.69|8.59|
> |MCPO|**300.25**|**100.28**|
>
>
> **"... choice of Atari games ..."**
>
> As mentioned in Sec. 4.4, they are based on the set of typical games introduced in a prior work (https://arxiv.org/pdf/1312.5602.pdf) and we tried to replace an easy game (e.g Pong) with a more challenging one (randomly chosen Gopher). Due to compute limit, we could not run for more games. We cannot guarantee our method to always perform well in all games, but we believe on average, our method shows strong performance.  To confirm the consistently good performance of our method, we will test with additional 3 games (Qbert-the final one in the set of typical games + another 2 randomly chosen games:  BankHeist, IceHockey). Please see Appendix Fig. 4 for the learning curves of these additional games.
>
> **"... Table 2 without seeing the learning curves ..."**
>
> Due to the space limit, we report Table 2 as a summary of performance. As shown in Table 2, the final performance gap between our methods and others is clear in hard problems like HalfCheetah, Ant, Humanoid, HumannoidStandup (Cohen effect size>0.5) while in other problems, we show comparable results.  We also reported full learning curves in Appendix Fig. 7 in the old version (Fig. 9 in the current version).
>
> **"It is better to show clear improvement on two benchmarks rather than 5 different ones without a clear conclusion"**
>
> Given the above explanation, it is clearer that our method shows strong performance in all tasks. We deem that consistent performance is very important because erratically big improvement in 1 or 2 benchmarks can be the result of task-specific tuning/overfitting. That's why we tried to examine the models in a wide range of tasks and are more interested in achieving consistently high performance.
>
> **"... the choice of $\beta$ ..."**
>
> We want to make it clear that $\beta$  is the common coefficient for both KL terms in Eq. 1. To differently weigh the two KL terms, we already have coefficient $\alpha$ designed as you suggested (with $\alpha$ and 1-$\alpha$). So $\alpha$ is the relative weight between 2 KL terms while $\beta$  is the general weight between trust-regions and advantage objective. The intuition for $\beta$ is if the current policy is too far from the old policy, $\beta$  should be high ($\beta$  can even >1) and vice versa. The question is how to detect if the policy is too far? If you just measure the "distance" between the current and the old policy, you will need to define a threshold, which requires additional hyperparameter tuning.  Instead, we make use of the virtual policy to create a "soft" threshold as explained in Eq. 2.

---

### Author Response · Authors · 2021-11-18
**General response**

Dear Reviewers,

Thank you for your thoughtful and constructive feedback.  Many of your comments are appropriate and based on them, we have revised our paper with changes as follows:
- Revise minor writing errors
- Add related works
- Add more experimental results (more baselines, more games, more iterations)

Please find the details in the individual responses.

However, there remain misunderstandings that may prevent a correct evaluation of our paper. We will address these in a detailed response to each reviewer. We hope that our responses will address your concerns and clarify misunderstandings. Please consider increasing your score if you find our responses valid.

---

### Decision · Program_Chairs · 2022-01-20

**Decision:**

Reject

**Comment:**

Two trust region constrained optimization for policy gradient RL, where the second trust region is based on a virtual policy built from a memory buffer and using an attention mechanism to combine prior policies. The reviewers agree that the paper is well written, the idea is novel, and the paper is extensively evaluated. The authors are commended for running the additional baselines during the rebuttal period.

However, the paper still contains some shortcomings, specifically, the results are somewhat inconclusive even after the rebuttal. While it is not expected that the method wins across the board, it is important to provide an analysis of the limitations of the method. When is the algorithm appropriate to use, and when is it not?

To make the paper stronger, in the next version of the paper should:
- move the theory in the main text (Appendix C).
- provide the analysis of the algorithm and its limitations.